# CD73 Promotes Tumor Progression in Patients with Esophageal Squamous Cell Carcinoma

**DOI:** 10.3390/cancers13163982

**Published:** 2021-08-09

**Authors:** Yen-Hao Chen, Hung-I Lu, Chien-Ming Lo, Shau-Hsuan Li

**Affiliations:** 1Department of Hematology-Oncology, Kaohsiung Chang Gung Memorial Hospital and Chang Gung University College of Medicine, Kaohsiung 833, Taiwan; alex2999@cgmh.org.tw; 2Department of Nursing, Meiho University, Pingtung 912, Taiwan; 3School of Medicine, Chung Shan Medical University, Taichung 402, Taiwan; 4Department of Thoracic & Cardiovascular Surgery, Kaohsiung Chang Gung Memorial Hospital and Chang Gung University College of Medicine, Kaohsiung 833, Taiwan; luhungi@cgmh.org.tw (H.-I.L.); t123207424@cgmh.org.tw (C.-M.L.)

**Keywords:** esophageal cancer, CD73, squamous cell carcinoma, epithelial–mesenchymal transition, esophagectomy

## Abstract

**Simple Summary:**

The immune system plays important roles in antitumor activities. However, increasing evidence shows that tumor cells develop several mechanisms to escape the immune attack, resulting in immunosuppression. One of the most important immunosuppressive pathways is the CD73-adenosinergic pathway. In addition, this pathway participates in the development of cancer, including tumor cell proliferation, angiogenesis, and anti-inflammation mechanisms. Moreover, CD73 can mediate the invasion and metastasis of tumor cells via the regulation of cell interactions with the extracellular matrix components. Therefore, overcoming immunosuppression to restore the antitumor functions of T cells may be explored as a potential treatment strategy. Overexpression of CD73 promotes the malignant properties of cancers and is associated with specific clinical characteristics and worse prognosis in many types of cancers. The current study is the first to investigate the role of CD73 in determining the clinical outcomes of patients with esophageal squamous cell carcinoma.

**Abstract:**

Cluster of differentiation (CD)-73 plays pivotal roles in the regulation of immune reactions via the production of extracellular adenosine, and the overexpression of CD73 is associated with worse outcomes in several types of cancers. Here, we identified 167 esophageal squamous cell carcinoma (ESCC) patients who underwent esophagectomy, including 64 and 103 patients with high and low expression levels of CD73, respectively. Univariate and multivariate analyses showed high expression of CD73 was an independent prognostic factor for worse disease-free survival and overall survival. In addition, we selected another cohort consisting of 38 ESCC patients receiving nivolumab or pembrolizumab and found that treatment response and survival benefit to immunotherapy were strongly correlated with the expression levels of CD73/programmed death ligand 1. Moreover, the transwell assay revealed knockdown of CD73 in two ESCC cell lines, TE1 and KYSE30, exhibited significantly reduced abilities of cell invasion and migration. CD73 silencing also showed that the protein expression levels of CD73, vimentin, and snail were downregulated, while those of E-cadherin were upregulated in Western blotting. The findings of our study indicate CD73 may be an independent prognostic factor for ESCC patients who underwent esophagectomy. Furthermore, it may be associated with the patient responses to immunotherapy.

## 1. Introduction

Esophageal squamous cell carcinoma (ESCC) is the ninth leading cause of cancer-related mortality in Taiwan [1]. Most patients are diagnosed with locally advanced status as no obvious symptoms or signs are observed in the early stage of this disease. Despite significant improvements in the surgical techniques, chemotherapy, radiotherapy, and immunotherapy used for treatment, the outcomes of the patients with ESCC remain poor [2,3]. The 5-year survival rate is only around 15–20% [4,5]. Therefore, it is important to identify the key regulators of signal pathways involved in tumor progression to overcome the resistance to cancer treatment in patients with ESCC.

The immune system plays important roles in antitumor activities as the innate and adaptive immune systems recognize and remove abnormal cells, including tumor cells. However, increasing evidence shows that tumor cells develop several mechanisms to escape the immune attack, resulting in immunosuppression and pro-angiogenic activity to promote the onset and progression of cancer [6,7,8,9]. Therefore, overcoming immunosuppression to restore the antitumor functions of T cells may be explored as a potential treatment strategy. Significant developments have been made in the past decade in enhancing our understanding of the interactions between the cancer cells and the immune system. Immune checkpoint blockers (ICBs), monoclonal antibodies targeting cytotoxic T lymphocyte antigen 4 (CTLA4), programmed cell death 1 (PD-1), and programmed cell death ligand 1 (PD-L1) have been approved for the clinical management of cancer based on a series of phase III, randomized controlled trials [10]. Recently, immunotherapy has been approved for patients with ESCC according to the results of three phase III randomized clinical trials, including the KEYNOTE-181, KEYNOTE-590 (pembrolizumab), and ATTRACTION-3 (nivolumab) studies [11,12,13]. However, these medications are not effective for all patients, with some patients still exhibiting certain resistance to treatment. Therefore, it is necessary identify the mechanisms by which cancer cells escape the immune system to improve the clinical outcomes of the use of ICBs for the treatment of cancer.

One of the most important immunosuppressive pathways is the CD73-adenosinergic pathway [14,15]. This purinergic signaling pathway, a crucial part of the tumor microenvironment (TME), is considered to play important roles in the immune escape and cancer progression mechanisms via the stimulated release of extracellular ATP, ADP, and adenosine [16,17]. CD73, encoded by the ecto-5′-nucleotidase (*NT5E*) gene, is a 70 kD glycosylphosphatidylinositol-anchored cell membrane protein that plays an important role in the adenosinergic pathway. Extracellular ATP/AMP is converted into adenosine and phosphate by CD73. Adenosine is a major molecule involved in the suppression of antitumor T cell functions [18,19,20,21]. In addition, adenosine participates in the development of cancer, including tumor cell proliferation, angiogenesis, and anti-inflammation mechanisms [22,23]. Moreover, CD73 can mediate the invasion and metastasis of tumor cells via the regulation of cell interactions with extracellular matrix components, such as fibronectin and laminin [24,25].

Overexpression of CD73 promotes the malignant properties of cancers, such as proliferation, invasion, migration, adhesion, and metastasis, and it is associated with specific clinical characteristics and worse prognosis in many types of human cancers, including melanoma, leukemia, pancreatic cancer, triple-negative breast cancer, thyroid cancer, gastric cancer, prostate cancer, colon cancer, and ovarian cancer [26,27,28,29,30,31,32,33,34,35]. However, the specific function of CD73 in the progression of ESCC remains unclear. Therefore, the aim of this study was to elucidate the role of CD73 in determining the clinical outcomes of patients with ESCC who underwent esophagectomy.

## 2. Materials and Methods

### 2.1. Patient Selection

We retrospectively reviewed patients with ESCC who underwent esophagectomy between January 2001 and December 2015 at the Kaohsiung Chang Gung Memorial Hospital. First, we only included the patients with ESCC who underwent esophagectomy as curative treatment, while those patients who underwent preoperative chemotherapy, radiotherapy, or chemoradiotherapy as initial treatment were excluded from this study. Second, patients with other types of cancer, such as adenocarcinoma or small cell carcinoma, were also excluded from this study. We subsequently also excluded the cases in which a second primary malignancy was diagnosed within five years of primary ESCC. Patients aged <18 years were also excluded. Finally, 167 patients were identified. Each patient underwent chest computed tomography, endoscopic ultrasonography, and positron emission tomography scans to determine their clinical stage at diagnosis, and pathological staging was performed according to the 8th edition of the American Joint Committee on Cancer staging system [36].

We also enrolled patients with ESCC who received ICBs (including pembrolizumab and nivolumab) as second-line and later-line treatment in our institute between July 2020 and June 2021. Thirty-eight patients with ESCC were present in this cohort.

### 2.2. Immunohistochemistry

Immunohistochemical (IHC) staining was performed on the slides (4 μm) of formalin-fixed, paraffin-embedded tissue sections. First, the sections were deparaffinized by incubating them in a dry oven at 60 °C for 1 h. Then, antigen retrieval was done using 10 mM citrate buffer (pH 6.0), followed by incubation in a hot water bath at 95 °C for 20 min, and peroxidase blocking using 0.3% hydrogen peroxide for 5 min. Then, a primary antibody against CD73 (HPA017357, 1:1000; Sigma, Burlington, USA) was allowed to react with the sections. Later, a ready-to-use visualization reagent consisting of a goat secondary antibody was also added to the sections and allowed to react. The tissue sections were then incubated with a polymer for 8 min, followed by staining with 3,3′-diaminobenzidine for 10 min, and counterstained with hematoxylin. The negative control group samples were stained using an identical procedure, while a slide of the human testis was used as the positive control. The slides were scored by two pathologists (Chao-Cheng Huang and Wan-Ting Huang) who were blinded to the clinicopathological features or prognosis of the patients. A semi-quantitative immunoreactive score (IRS) was obtained to determine the expression levels of CD73 [37]. The IRS was calculated by multiplying the staining intensity, including the percentage of positively stained cells (0: no staining; 1: <10% of the cells; 2: 11–50%; 3: 51–80%; and 4: >81%) and histological grade (0, no staining; 1, weak staining; 2, moderate staining; and 3, strong staining). A specimen with a sum score >6 was considered to be positively stained (Figure 1).

Immunohistochemical staining of PD-L1 was performed according to the method described above. Each IHC run contained a positive control and a negative antibody control, and the PD-L1 IHC 22C3 pharmDx assay (Dako, Carpinteria, CA, USA) was used. The specimens were incubated with anti-human PD-L1 monoclonal mouse antibody (#29122, 1:50; Cell Signaling, Danvers, Massachusetts, USA), and the monoclonal mouse control IgG antibody and human placenta were used as the negative and positive controls, respectively. The expression levels of PD-L1 were determined according to the combined positive score (CPS), which is defined as the total number of tumor cells and immune cells (lymphocytes and macrophages) stained with PD-L1 divided by the total number of viable tumor cells, multiplied by 100. High PD-L1 expression was defined by a CPS ≥ 10 [13].

### 2.3. Cell Lines and Culture

ESCC cell lines, TE1 and KYSE30, were used in this study. KYSE30 was obtained from Public Health England, and TE1 was purchased from the Cell Resource Center for Biomedical Research Institute of Development, Aging and Cancer (Tohoku University, Sendai, Japan). These cell lines were cultured in Dulbecco’s modified Eagle’s medium (DMEM) nutrient mixture F-12 (Sigma–Aldrich). All culture media contained 10% fetal bovine serum (FBS). The cells were then cultured at 37 °C.

### 2.4. Knockdown of CD73 by Lentiviral Transduction

CD73 shRNA and control shRNA plasmids were purchased from the National RNAi Core Facility of Academia Sinica (Taipei, Taiwan). The sequences of shRNAs were as follows: 5′-GCCACTGTCAACATCCTCATA-3′ for CD73 and 5′-GCGGTTGCCAAGAGGTTCCAT-3′ for control. CD73 shRNA-expressing virus was prepared by co-transfection of the plasmid containing a shRNA cloned, pCMV-ΔR8.91, and pMD.G into HEK-293T cells by using TurboFect transfection reagent according to the procedure. Cultures of virus-infected TE1 and KYSE30 cells were selected with puromycin antibiotic for knockdown strains.

### 2.5. Migration and Invasion Assays

Transwell inserts (pore size, 8 mm; Corning, Glendale, AZ, USA) were used to evaluate the cell migration, and Matrigel (BD Biosciences, San Jose, CA, USA)-coated porous filters were used to examine the cell invasion abilities. Cells (1 × 10^4^) in 200 mL DMEM containing 10% FBS were seeded into these inserts, and 600 mL was added to the lower part of the well. The cells were incubated for 24 h. Then, the cells on the upper side of the membrane were wiped, while those moving to the other side of the filters were stained with crystal violet and counted using a microscope in three randomly selected fields. The independent experiments were repeated thrice.

### 2.6. Western Blotting Analysis

Whole-cell lysates of CD73-shRNA-treated cells were extracted with 300 μL of radioimmunoprecipitation assay (RIPA) buffer (50 mM Tris, 150 mM sodium chloride (NaCl), 1% NP40, 0.5% sodium deoxycholate, and 0.1% sodium dodecyl sulfate (SDS)) and subjected to Western blotting analysis. The membranes were then incubated with polyclonal antibodies against CD73 (ab91086, 1:1000; Abcam, Cambridge, UK), E-cadherin (GTX124178, 1:5000; Genetex, Irvine, CA, USA), vimentin (ab92547, 1:1000; Abcam, Cambridge, UK), snail (#3879, 1:1000; Cell Signaling, Danvers, MA, USA), and β-actin (A5441, 1:10000; Sigma-Aldrich, St. Louis, Missouri, USA). Horseradish peroxidase-conjugated anti-rabbit secondary antibody was added to detect the primary antibodies, and the blots were developed using a chemiluminescence system (Pierce). All resolved protein bands were developed using the Western Lightning Chemiluminescence Reagent Plus system (Amersham Biosciences). All experiments were repeated at least three times, with similar results. The whole Western Blot figures can be found in the Appendix A.

### 2.7. Statistical Analysis

Baseline characteristics were expressed as numbers and percentages. The chi-square test was used to compare the categorical variables. Disease-free survival (DFS) was calculated from the time of surgery to the time of tumor recurrence or death from any cause without evidence of recurrence. Overall survival (OS) was defined as the duration from the time of ESCC diagnosis to death or the time of last living contact. The effects of variables on DFS and OS were determined using the univariate and multivariate Cox proportional hazards models, and the Kaplan–Meier method and log-rank test were performed for survival curve analysis. All statistical analyses were performed using the SPSS software v.22 (International Business Machines Corp., Armonk, NY, USA). A two-tailed *p*-value of < 0.05 was considered to indicate statistical significance in all analyses.

### 2.8. Ethics Statement

This study was approved by the Chang Gung Medical Foundation Institutional Review Board (202002185B0). All procedures involving human subjects were performed in accordance with the ethical standards of the Institutional Research Committee and the World Medical Association Declaration of Helsinki.

## 3. Results

### 3.1. Patient Characteristics

A total of 167 patients with ESCC who underwent esophagectomy and met the inclusion/exclusion criteria in our institute were identified, including 161 men and 6 women, with a mean age of 55 years (range: 29–81 years). The pathological tumor (T) status revealed 58 patients with T1 disease (34.7%), 33 patients with T2 disease (19.8%), 61 patients with T3 disease (36.5%), and 15 patients with T4 disease (9.0%). The pathological node (N) status showed N0 disease in 115 patients (68.8%), N1 disease in 32 patients (19.2%), N2 disease in 12 patients (7.2%), and N3 disease in 8 patients (4.8%). There were 53 patients (31.7%) with stage I, 60 patients (35.9%) with stage II, 33 patients (19.8%) with stage III, and 21 patients (12.6%) with stage IVA diseases. Primary tumor locations were found to be the upper third ESCC in 26 patients (15.6%), middle ESCC in 65 patients (38.9%), and lower third ESCC in 76 patients (45.5%). Analysis of tumor grade demonstrated that 19 patients (11.4%) were diagnosed with grade 1, 107 patients (64.1%) with grade 2, and 41 patients (24.5%) with grade 3 tumors. There was no statistical difference of age, sex, pathological N status, tumor location, and tumor grade between patients with high or low expression of CD73; however, patients with high expression of CD73 had higher percentage of advanced pathological T status and pathological tumor stage compared to those with low expression of CD73. At the time of analysis, the median periods of follow-up were 35.7 months for all 167 patients and 69.7 months for the 57 survivors, respectively. The clinicopathological characteristics of the patients are shown in Table 1.

### 3.2. CD73 Expression and Clinical Outcome

There were 64 patients (38.3%) with high expression levels of CD73 and 103 patients (61.7%) with low expression levels of CD73. In the analysis of DFS, there were no significant differences in the sex and tumor location in the univariate analysis. Better DFS was found in the patients aged <60 years (*p* = 0.046), low pathological T stage (T1–2) (*p* < 0.001), negative nodal metastasis (*p* < 0.001), low pathological tumor stage (stage I–II) (*p* < 0.001), and low tumor grade (grade 1–2) (*p* < 0.001). The 103 patients with low expression levels of CD73 had significantly superior DFS compared to the 64 patients with high expression levels of CD73 (not reached versus 12.3 months, *p* < 0.001, Figure 2A). According to a multivariate comparison, pathological T1–T2 (*p* = 0.006, hazard ratio (HR): 0.45, 95% confidence interval (CI): 0.27–0.80), negative nodal metastasis (*p* = 0.005, HR: 0.50, 95% CI: 0.30–0.81), and low expression of CD73 (*p* = 0.040, HR: 0.61, 95% CI: 0.37–0.97) were independent prognostic factors for superior DFS.

With respect to OS, sex and tumor grade were not statistically significant predictors of OS in the univariate analysis. Meanwhile, patients below 60 years old (*p* = 0.013), low pathological T stage (T1–2) (*p* < 0.001), negative nodal metastasis (*p* < 0.001), low pathological tumor stage (stage I–II) (*p* = 0.004), and low tumor grade (grade 1–2) (*p* < 0.001) were found to have superior OS. Better OS was observed in 103 patients with low expression levels of CD73 compared to the 64 patients with high expression levels of CD73 (66.0 months vs. 13.4 months, *p* < 0.001, Figure 2B). Moreover, the multivariate analyses showed that pathological stage I and II (*p* < 0.001, HR: 0.38, 95% CI: 0.25–0.57) and low expression levels of CD73 (*p* < 0.001, HR: 0.47, 95% CI: 0.32–0.69) were independent predictive factors of better OS. The univariate and multivariate analyses of DFS and OS in the 167 patients with ESCC are shown in Table 2 and Table 3, respectively.

### 3.3. Correlation between Immunotherapy and the Expression Levels of CD73/PD-L1

We enrolled 38 patients who received pembrolizumab or nivolumab as second-line or later-line treatment for ESCC. The clinicopathological characteristics of the patients are shown in Table 4. Based on the expression levels of CD73 and PD-L1 (Figure 3), these patients were divided into four groups: high expression levels of both CD73 and PD-L1 (Group A), high expression levels of CD73/low expression levels of PD-L1 (Group B), low expression levels of CD73/high expression levels of PD-L1 (Group C), and low expression levels of both CD73 and PD-L1 (Group D). Treatment responses to immunotherapy were found to be associated with the expression levels of both CD73 and PD-L1. For 10 patients with partial response (PR), a higher PR rate (60%) was noted in the group C; for 20 patients who had disease progression (PD), a higher percentage of PD (45%) was found in group B. More than half of the patients (60%) with low expression levels of CD73/high expression levels of PD-L1 exhibited PR to immunotherapy; however, PD was up to 81.8% in the high expression levels of CD73/low expression levels of PD-L1 group. The response to immunotherapy was strongly correlated with the expression levels of CD73/PD-L1 (*p* = 0.010), and there was a statistical difference in treatment response between group B and group C (*p* = 0.004). Moreover, the median PFS in group A, group B, group C, and group D were 9.4, 1.4, 5.0, and 1.7 months, respectively (*p* = 0.003); patients in group C had better PFS than those in group B (*p* = 0.003). In addition, the median OS was not reached in group A, 7.7 months in group B, 13.4 months in group C, and 12.3 months in group D (*p* = 0.012); a superior OS was found for group C compared to group B (*p* = 0.034). The treatment outcome of immunotherapy was shown in Table 5.

### 3.4. CD73 Silencing Decreases the Migration and Invasion Abilities of ESCC Cells

In our study, two ESCC cell lines, TE1 and KYSE30, were used to test the effect of CD73 on tumor cell migration and invasion. First, we determined the cellular motility of the ESCC cells treated with CD73 shRNA using a transwell assay. The results of the transwell assay revealed that the cells treated with CD73-shRNA significantly reduced the number of invaded and migrated cells compared to the cells with CD73-shControl treatment (Figure 4). The observation showed that, at least in TE1 and KYSE30 cell lines, CD73 silencing could suppress the motility of ESCC cells. Moreover, Western blotting analyses were performed to determine the expression levels of CD73 and epithelial–mesenchymal transition (EMT). Our data showed that the protein expression levels of CD73, vimentin, and snail were downregulated, while that of E-cadherin was upregulated in the CD73 shRNA-treated cell lines compared to the control cells (Figure 5). Collectively, these data revealed that the expression of CD73 is involved in the motility of ESCC cells.

## 4. Discussion

The TME is the primary location for the interactions of the immune cells and tumor cells. TME plays a crucial role in the treatment of cancer, especially causing resistance to treatment [38]. Specific oncogenes may enhance malignant properties of tumor cells via the modulation of TME, which may lead to tumor progression [39]. Purinergic signaling is an important component of TME that is responsible for the communication of cells in the physical and pathological settings [16,17]. The complex network of purinergic signaling events are involved in the immune escape, promotion of tumor cell growth, accelerated migration and invasion, and metastasis of cancer cells [39,40]. CD73 is a membrane-bound enzyme that catalyzes the conversion of extracellular AMP to adenosine and is responsible for the modulation of the immune system. The adenosinergic pathway plays an important role in the regulation of anti-tumor T cell responses. The functions of CD73 include the limitation of anti-tumor T cell expansion, homing of tumors, and induction of immunosuppression and cancer cell survival [41,42]. In addition, CD73 is also associated with mechanisms related to carcinogenesis, escape from apoptosis, and resistance to chemotherapy [43,44,45]. CD73 overexpression has been observed in several types of cancer, such as gastric, pancreatic, and triple-negative breast cancers, and it is regarded as an independent prognostic factor [31,46,47]. The findings of our study also confirmed the prognostic role of CD73 in ESCC.

Epithelial–mesenchymal transition (EMT) plays an important role in all stages of tumor progression, including initiation, proliferation, migration, invasion, metastasis, and resistance to anti-tumor treatment [48]. Several studies have shown a correlation between EMT and CD73. Petruk et al. reported that CD73 facilitates progression in triple-negative breast cancer [49]. CD73 silencing resulted in increased expression of E-cadherin and decreased expression of vimentin in vitro and in vivo, indicating maintenance of a more epithelial phenotype [49]. Overexpression of CD73 was regarded as an independently poor prognostic indicator for tumor recurrence and overall survival in hepatocellular carcinoma [50]. In addition, CD73 knockdown dramatically decreased the expression of N-cadherin, vimentin, and twist and increased the expression of E-cadherin, and opposite results were observed when CD73 was overexpressed [50]. Xu et al. also reported that CD73 modulates EMT process in gastric cancer, including opposite results in CD73-overexprressed cells and CD73-knockdown cells in vitro [47]. In our study, CD73 silencing decreased tumor cell migration, invasion, and EMT process in ESCC. Here, we provide the first evidence that CD73 may be associated with EMT in ESCC.

CD73 promotes tumor progression via adenosine metabolism, including the inhibition of anti-tumor immune responses and induction of angiogenesis, and it has recently emerged as a promising target for novel immunotherapy. Extensive research has shown that CD73 small molecule inhibitors could decrease tumor cell progression and increase survival in several preclinical tumor mouse models, such as melanoma, breast cancer, and prostate cancer [51]. Recently, immune checkpoint blockades have been approved for cancer treatment in several cancer types, including first-line, second-line, or later lines. However, the expression levels of CD73 of the tumor cells may attenuate the immune response evoked by anti-PD-1 treatment, resulting in poor response and clinical outcome [52]. Although the immune checkpoint blockade is still effective in certain cancer patients, many patients do not respond to immunotherapy or have a long-term duration of response. Therefore, the combination of the CD73 inhibitor and the immune checkpoint blockade, such as CTLA-4- and PD-1 monoclonal antibody, could enhance the anti-tumor effects of these agents compared to monotherapy in several murine tumor models [14]. Furthermore, several CD73 blockades, including selective small molecule inhibitors and anti-CD73 monoclonal antibody, are being tested in several early phase clinical trials [53,54].

In a phase III study of advanced esophageal cancer, KEYNOTE-181, patients with CPS ≥ 10 who received pembrolizumab had a better response rate and OS compared to those who received chemotherapy [12]. Another phase III study that focused on ESCC, KEYNOTE-590, showed that ESCC patients with CPS ≥ 10 who received pembrolizumab plus chemotherapy had a superior response rate, PFS, and OS than those with chemotherapy alone; however, there were no statistical differences of PFS and OS between patients with CPS < 10 who received immunotherapy plus chemotherapy and chemotherapy alone [13]. These two phase III studies both indicated that CPS ≥ 10 may be a predictive factor of better outcome to immunotherapy in esophageal cancer patients. In addition, previous research and our study also confirmed that CD73 is a poor prognostic factor in several cancer types, including ESCC. In our study, a better response to nivolumab/pembrolizumab, two kinds of anti-PD-1 monoclonal antibodies, was found in patients with low expression levels of CD73 and high expression levels of PD-L1. In contrast, disease progression was found in more than 80% of patients with high expression levels of CD73 and low expression levels of PD-L1. Moreover, better PFS and OS were found in patients with low expression levels of CD73 and high expression levels of PD-L1 compared to those with high expression levels of CD73 and low expression levels of PD-L1. Therefore, the results of our study confirmed the mechanism mentioned above.

This study has several limitations. First, this was a single-institution study with a relatively small sample size. Second, there were only six female patients in our study, which may have contributed to some bias in gender-based differences in the estimation of survival rates. However, to the best of our knowledge, our study is the first to investigate the role of CD73 in the pathogenesis of ESCC. Much larger prospective human and animal studies are needed to validate these findings.

## 5. Conclusions

Our study shows that CD73 may act as an independent prognostic factor for patients with ESCC who underwent esophagectomy and may also influence the patient responses to immunotherapy.

## Figures and Tables

**Figure 1 cancers-13-03982-f001:**
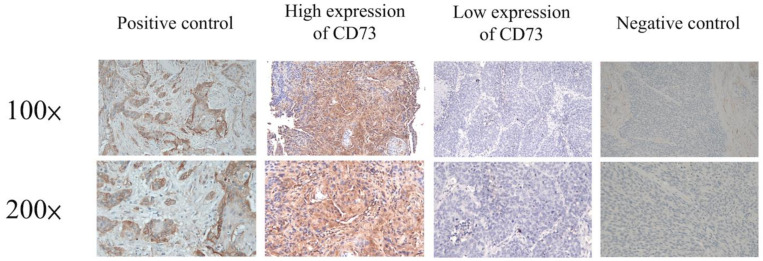
Results of the immunohistochemical analysis of cluster of differentiation (CD)-73 in patients with esophageal squamous cell carcinoma (ESCC).

**Figure 2 cancers-13-03982-f002:**
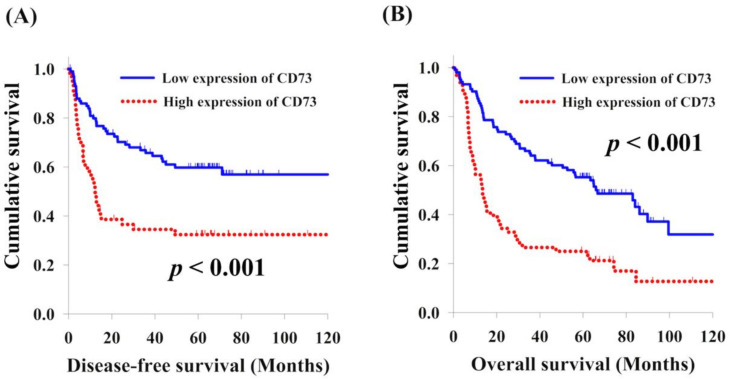
Comparison of the Kaplan–Meier curves of the patients with ESCC with high and low expression levels of CD73. (**A**) Disease-free survival and (**B**) overall survival.

**Figure 3 cancers-13-03982-f003:**
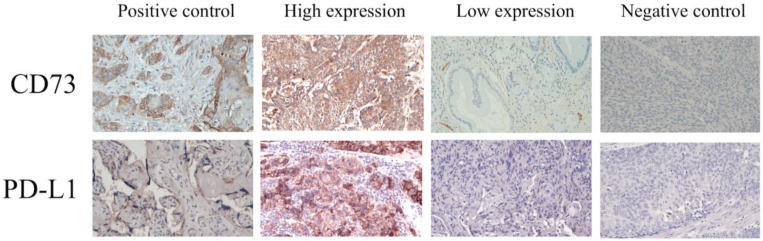
Results of the immunohistochemical analysis of CD73 and programmed death ligand 1 (PD-L1) in the patients with ESCC.

**Figure 4 cancers-13-03982-f004:**
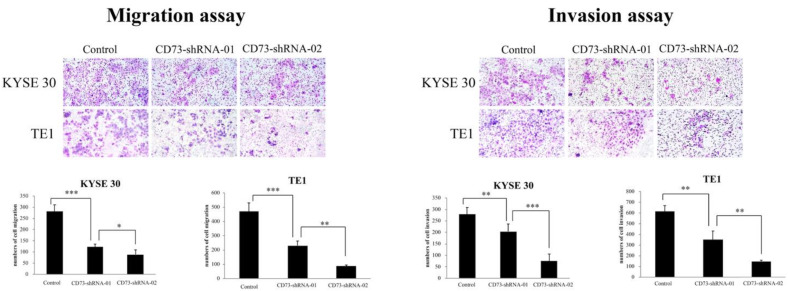
Transwell migration and invasion assays using TE1 and KYSE30 cell lines treated with the CD73-short hairpin RNA (shRNA). Columns, mean; bars, standard deviation. Significant difference: ** p* < 0.05, ** *p* < 0.01, and *** *p* < 0.001.

**Figure 5 cancers-13-03982-f005:**
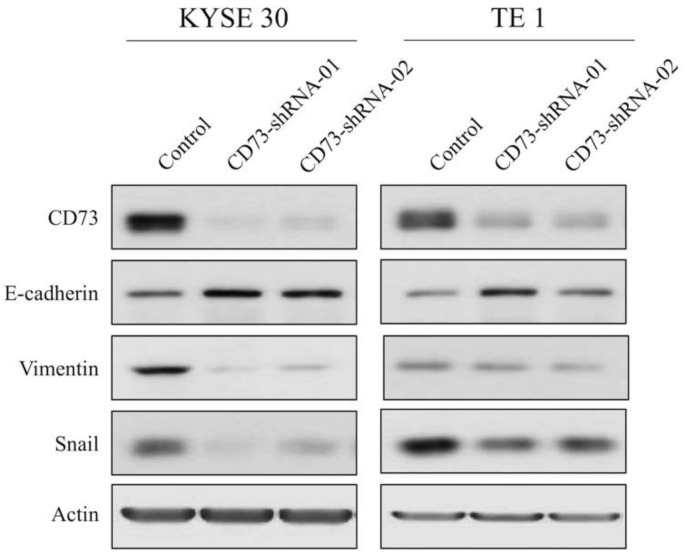
Western blotting analysis of the expression levels of CD73 and the downstream signaling pathways in the TE1 and KYSE30 cell lines. The protein expression profiles of CD73, E-cadherin, vimentin, and snail were examined in the presence or absence of CD73-shRNA treatment in the ESCC cells via Western blotting.

**Table 1 cancers-13-03982-t001:** Characteristics of 167 patients with esophageal squamous cell carcinoma receiving surgical resection.

Characteristics	High Expression of CD73 (*n* = 64)	Low Expression of CD73 (*n* = 103)	*p*-Value
Age (years)	55 years old (39–77)	58 years old (29–81)	0.13
Sex			
Male	2 (3.1%)	4 (3.9%)	0.80
Female	62 (96.9%)	99 (96.1%)	
Pathological T status			
1	49 (47.6%)	9 (14.1%)	*p* < 0.001 *
2	23 (22.3%)	10 (15.6%)	
3	26 (25.2%)	35 (54.7%)	
4	5 (4.9%)	10 (15.6%)	
Pathological N status			
0	77 (74.8%)	38 (59.4%)	0.16
1	16 (15.5%)	16 (25.0%)	
2	7 (6.8%)	5 (7.8%)	
3	3 (2.9%)	5 (7.8%)	
Pathological tumor stage			
I	46 (44.7%)	8 (12.5%)	*p* < 0.001 *
II	33 (32.0%)	26 (40.6%)	
III	17 (16.5%)	16 (25.0%)	
IVA	7 (6.8%)	14 (21.9%)	
Location			
Upper	19 (18.4%)	7 (10.9%)	0.39
Middle	40 (38.8%)	25 (39.1%)	
Lower	44 (42.8%)	32 (50.0%)	
Grade			
1	14 (13.6%)	5 (7.8%)	0.05
2	70 (68.0%)	37 (57.8%)	
3	19 (18.4%)	22 (34.4%)	

* Statistically significant.

**Table 2 cancers-13-03982-t002:** Univariate and multivariate analysis of disease-free survival (DFS) in 167 patients with esophageal squamous cell carcinoma receiving surgical resection.

Characteristics	No. of Patients	Univariate Analysis			Multivariate Analysis	
		Median DFS (Months)	HR (95% CI)	*p*-Value	HR (95% CI)	*p*-Value
Age						
<60 years	106 (63.5%)	NR	0.64 (0.41–0.99)	0.046 *	0.79 (0.50–1.26)	0.32
≥60 years	61 (36.5%)	24.8				
Sex						
Male	161 (96.4%)	49.3		0.34		
Female	6 (3.6%)	NR	0.51 (0.13–2.10)		0.43 (0.10–1.79)	0.25
Pathological T status						
1 + 2	91 (54.5%)	NR	0.32 (0.20–0.50)	<0.001 *	0.45 (0.27–0.80)	0.006 *
3 + 4	76 (45.5%)	12.2				
Pathological N status						
0	115 (68.9%)	NR	0.36 (0.23–0.56)	<0.001 *	0.50 (0.30–0.81)	0.005 *
1 + 2 + 3	52 (31.1%)	12.3				
Pathological tumor stage						
I + II	113 (67.6%)	NR	0.34 (0.22–0.53)	<0.001 *	0.77 (0.32–1.87)	0.56
III + IVA	54 (32.4%)	9.8				
Location						
Upper + Middle	91 (54.5%)	49.3	0.96 (0.62–1.49)	0.86	0.80 (0.51–1.27)	0.35
Lower	76 (45.5%)	43.5				
Grade						
1 + 2	128 (76.6%)	NR	0.43 (0.27–0.69)	<0.001 *	0.62 (0.38–1.00)	0.05
3	39 (23.4%)	12.8				
CD73 expression						
High	64 (38.3%)	12.3		<0.001 *		
Low	103 (61.7%)	NR	0.43 (0.28–0.68)		0.61 (0.37–0.97)	0.040 *

NR: not reach; HR: hazard ratio; CI: confidence interval; * Statistically significant.

**Table 3 cancers-13-03982-t003:** Univariate and multivariate analysis of overall survival (OS) in 167 patients with esophageal squamous cell carcinoma receiving surgical resection.

Characteristics	No. of Patients	Univariate Analysis			Multivariate Analysis	
		Median OS (Months)	HR (95% CI)	*p*-Value	HR (95% CI)	*p*-Value
Age						
<60 years	106 (63.5%)	63.1	0.62 (0.43-0.91)	0.013 *	0.80 (0.54–1.19)	0.80
≥60 years	61 (36.5%)	26.3				
Sex						
Male	161 (96.4%)	33.1		0.20		
Female	6 (3.6%)	NR	0.41 (0.10–1.67)		0.36 (0.09–1.49)	0.16
Pathological T status						
1 + 2	91 (54.5%)	74.2	0.42 (0.12–0.61)	<0.001 *	0.69 (0.43–1.09)	0.11
3 + 4	76 (45.5%)	13.8				
Pathological N status						
0	115 (68.9%)	65.0	0.38 (0.26–0.55)	<0.001 *	0.62 (0.30–1.28)	0.19
1 + 2 + 3	52 (31.1%)	11.1				
Pathological tumor stage						
I + II	113 (67.6%)	65.0	0.36 (0.24–0.52)	<0.001 *	0.38 (0.25–0.57)	<0.001 *
III + IVA	54 (32.4%)	10.6				
Location						
Upper + Middle	91 (54.5%)	35.2		0.81		
Lower	76 (45.5%)	35.7	0.96 (0.66–1.39)		0.87 (0.59–1.28)	0.48
Grade						
1 + 2	128 (76.6%)	56.3	0.55 (0.36–0.83)	0.004 *	0.74 (0.47–1.14)	0.17
3	39 (23.4%)	15.2				
CD73 expression						
High	64 (38.3%)	13.4		<0.001 *		
Low	103 (61.7%)	66.0	0.41 (0.28–0.59)		0.47 (0.32–0.69)	<0.001 *

NR: not reach; HR: hazard ratio; CI: confidence interval; * Statistically significant.

**Table 4 cancers-13-03982-t004:** Characteristics of 38 ESCC patients who received pembrolizumab/nivolumab as second-line or later treatment.

Characteristics	Patient Numbers (%)
Age (years)	59 years old (42–71)
Sex	
Male	38 (100.0%)
Clinical T status	
2	8 (21.0%)
3	13 (34.3%)
4	17 (44.7%)
Clinical N status	
0	5 (13.3%)
1	11 (28.9%)
2	11 (28.9%)
3	11 (28.9%)
Clinical M status	
0	11 (28.9%)
1	27 (71.1%)
Clinical tumor stage	
III	5 (13.3%)
IV	33 (86.7%)
Location	
Upper	13 (34.3%)
Middle	14 (36.8%)
Lower	11 (28.9%)
Grade	
1	5 (13.3%)
2	27 (71.1%)
3	6 (15.6%)

**Table 5 cancers-13-03982-t005:** The correlation of treatment response to immunotherapy and expression of CD73/PD-L1 in 38 ESCC patients who received pembrolizumab/nivolumab.

Groups		Treatment Response							
		Partial Response (*n* = 10)	Stable Disease (*n* = 8)	Progressive Disease (*n* = 20)	*p*-Value	Progression-Free Survival (Months)	*p*-Value	Overall Survival (Months)	*p*-Value
A	CD73 high expression/PD-L1 high expression (*n* = 5)	1 (20.0%)	2 (40.0%)	2 (40.0%)	0.010 *	9.4	0.003 *	Not reach	0.012 *
B	CD73 high expression/PD-L1 low expression (*n* = 11)	1 (9.1%)	1 (9.1%)	9 (81.8%)		1.4		7.7	
C	CD73 low expression/PD-L1 high expression (*n* = 10)	6 (60.0%)	3 (30.0%)	1 (10.0%)		5.0		13.4	
D	CD73 low expression/PD-L1 low expression (*n* = 12)	1 (8.3%)	2 (16.7%)	9 (75.0%)		1.7		12.3	

* Statistically significant.

## Data Availability

Data are contained within the article.

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
