# Peer review of "CD73 Promotes Tumor Progression in Patients with Esophageal Squamous Cell Carcinoma"

_cancers, 2021, doi:10.3390/cancers13163982_

Round 1
Reviewer 1 Report
The present study shows that CD73 could be a prognostic factor for ESCC patients.
Here are my concerns.
Many references cited in the paper are not updated or not appropriated. For example data on 5-year survival rate is referred to Ref #4 published in 1996. This reference need to be updated. Ref 5, 14 need to be replaced with more appropriated references. Page 2 lane 85 ref #17 is not referred to CD73 and regulation of cell interactions with the extracellular matrix components. Page 2 lane 90 please include references on CD73 in human cancers (melanoma, leukemia are lacking). Page 10 lane 311 Petruk et al. is not numbered. Check carefully the references cited in the paper and replace them accordingly.
Page 1 lane 34 to 36, the sentence is not clear, please rewrite it.
Cells used in this paper have been treated with shRNA to silence CD73 gene expression. Which shRNA has been used? Treatment of cells with shRNA for CD73 is not described in the Material and Methods Section; please also add the relative control used for these experiments. More details should be provided in the methods section.
Importantly, regarding the results obtained in cells in which CD73 has been silenced, in my opinion stating “inhibition of CD73” is not appropriated. Here the authors do not use “inhibitors” of CD73. This should be corrected throughout the paper.
In figure 4 and figure 5 it is not clear what do shCD73-01 and shCD73-02 meant. Please clarify in the figure legends.
Resolution of Figure 4 is poor.
Silencing of CD73 in these two cell lines is associated with reduction of Vimentin and Snail and increase of E-Cadherin. These results are too preliminary to state that CD73 serves as trigger for EMT in ESCC.
The length of survival follow up for the cohorts assessed should be stated.
In Table 4 It is not clear if the p value is referred to all groups or to some of them only. Please clarify.
Data on the correlation between response to anti PD-1 agents and expression levels of CD73 and PD-L1 are very interesting. The clinical relevance of these results needs a further explanation.
Reviewer 2 Report
The authors investigated the role of CD73 expression in ESCC by the retrospective study with in vitro study. In this study, CD73 expression was found as a prognostic factor in patients with resectable ESCC, a predictive factor in patients with advanced ESCC treated by PD-1 inhibitor, and an aggressive factor in ESCC cell lines. Although this is an interesting and essential investigation, some revisions might be needed.
Major revision
- The authors examined CD 73 expression using formalin-fixed, paraffin-embedded tissue sections and used its semi-quantitative immunoreactive score. In the current manuscript, it is unclear which tissues (primary tumor, or metastatic tumor [e.g., lymph node metastasis) were used for the evaluation of CD 73 expression. I think both of primary tumor and lymph node metastasis were used for the analyses because of the description of negative nodal expression of CD73 (line 221). If so, the authors should document both types of CD 73 expression and used for analysis and discussion throughout the manuscript for clear understanding.
- In Table 2 and 3, the authors should document hazard ratio and p value regarding univariate analyses using Cox proportional hazards models. In addition, some factors with significance needed to be included for multivariate analyses.
Minor revision
- In Table 1, it is preferred that the clinicopathological factors are compared between the high CD73 expression group and the low CD 73 expression group.
- Line 244: a table for explaining the clinicopathological factors for patients receiving pembrolizumab/nivolumab are needed. In addition, it is interesting to know that progression-free survival and overall survival of patients receiving pembrolizumab/nivolumab according to CD73/PD-L1 status for knowing the proportion of patients with a durable response.
- Line 232: Figure 2A should be modified as Figure 2B.
- Line 190, 365: This is the retrospective study approved on 31 December 2020. Therefore, the description of “Informed consent was obtained from all subjects involved in the study” seems to be incorrect because the majority of the patients have already died before the study approval.
Round 2
Reviewer 1 Report
all the issues have been addressed.
